# MSAFlow: a Unified Approach for MSA Representation, Augmentation, and Family-based Protein Design

## Abstract

Multiple Sequence Alignments (MSAs) encode evolutionary information essential for protein structure prediction and functional design. However, high-quality MSAs require substantial computational resources for database searches, and homology search methods retrieve insufficient sequences for proteins with limited evolutionary relatives. While recent generative models have been proposed for MSA augmentation, they face challenges in capturing sequence dependencies while maintaining permutation invariance, and incur high memory costs due to quadratic complexity of self-attention on two-dimensional MSA representations. We present MSAFlow, a framework combining two components. First, we develop a generative autoencoder pairing a compressed AlphaFold3 (AF3) MSA representation with a conditional Statistical Flow Matching (SFM) decoder that models a family's sequence distribution while preserving permutation invariance. Second, we introduce a latent flow-matching model that generates MSA embeddings from a single sequence, enabling augmentation for orphan proteins. These components enable MSA representation, augmentation, and family-based design within a single framework. Evaluations demonstrate that MSAFlow achieves competitive performance on family-based protein design and MSA augmentation tasks, particularly for low-homology proteins. On CAMEO proteins, reconstructions from compressed MSA embeddings achieve structure prediction metrics (pLDDT 89.0, TM-score 0.86) approaching full MSAs (pLDDT 91.6, TM-score 0.89) while using 6.5% of the storage. For enzyme design with fewer than 20 training sequences, MSAFlow achieves 83-95% accuracy-uniqueness scores. MSAFlow is lightweight, fast, and memory-efficient, offering a versatile solution for diverse protein engineering tasks.

## 1 Introduction

Multiple Sequence Alignments (MSAs) collect homologous protein sequences that share evolutionary ancestry, providing fundamental information about protein evolution that plays crucial roles in downstream tasks such as structure prediction and family-based sequence design (Gong et al., 2025; Truong Jr & Bepler, 2023; Chen et al., 2024; Zhang et al., 2024a; Cao et al., 2025). These alignments represent evolutionary profiles that enable identification of conserved regions, such as key active site residues for enzymes, and evolutionary couplings that inform three-dimensional structure.

Conventional homology search tools such as HHBlits (Remmert et al., 2012), MMSeqs (Steinegger & Söding, 2017), and JackHMMER (Johnson et al., 2010) require substantial computational resources for obtaining high-quality MSAs. More critically, despite recent acceleration of MMSeqs2 with GPUs (Kallenborn et al., 2025), these methods retrieve insufficient sequences for low-homology and orphan proteins when evolutionary relatives are scarce in natural databases. This limitation motivates the development of tools that can generate MSAs and augment limited evolutionary data**, which are** essential for expanding protein structure prediction and functional analysis capabilities. Recent work has partially addressed MSA augmentation challenges through several approaches. Dense Homology Retriever (DHR) (Hong et al., 2024) leverages embeddings from protein language models to identify homologous sequences more efficiently and with greater sensitivity. Additional models, including MSAGenerator (Zhang et al., 2024b), MSAGPT (Chen et al., 2024), and EvoDiff (Alamdari

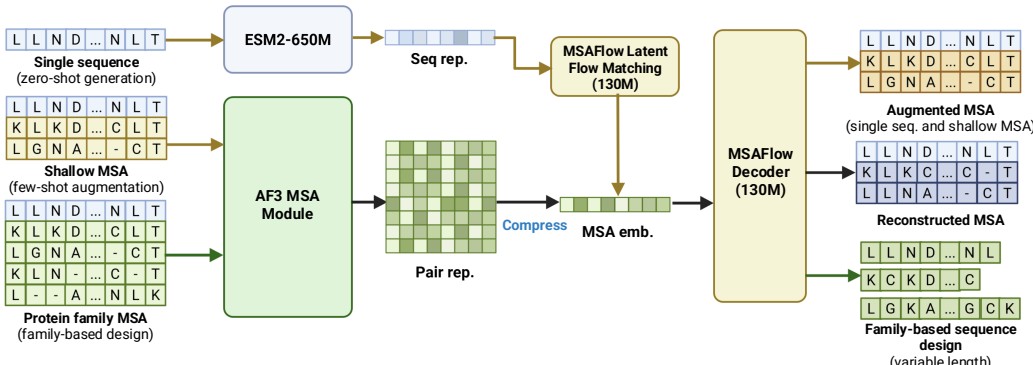

Figure 1: **General framework of MSAFlow.** Our approach supports three complementary pathways: (1) zero-shot generation from a single sequence using ESM2 embeddings, (2) few-shot augmentation of shallow MSAs, and (3) family-based design given MSAs embedded through the AF3 MSA Module and reconstructed through MSAFlow Decoder. All pathways leverage the latent flow-matching and decoder architecture to generate augmented or compressed MSAs, enabling both the enhancement of limited evolutionary information and the efficient representation of deep alignments.

et al.), have emerged, employing autoregressive and discrete diffusion frameworks, respectively to generate MSAs. While these methods show promise, they face architectural limitations in capturing distributional information while preserving permutation invariance. These methods typically utilize 2D positional encodings to represent row-wise and column-wise information in MSAs. This design incurs substantial memory costs due to the $O(N^2)$ space complexity of self-attention operations, which is further exacerbated by the 2D nature of MSAs. Additionally, methods like MSAGPT lack true permutation invariance due to left-to-right autoregressive decoding that introduces artificial sequential dependencies.

Beyond augmentation, improved generative models for MSAs that capture higher-order evolutionary patterns can serve as tools for guiding functional protein design. Potts models (Seemayer et al., 2014) pre-defined graphical models restricted to pairwise couplings. ProfileBFN (Gong et al., 2025) collapses sequence information into position-wise profiles that obscure higher-order dependencies, and methods such as MSA Transformer Rao et al. (2021) and EvoDiff (Alamdari et al.) flatten MSAs into 2D grids rather than explicitly modeling distributions over sequence space. These limitations motivate the development of a generative framework that can better approximate the underlying sequence distribution within an MSA without imposing strong assumptions.

To address these limitations, we introduce MSAFlow, a lightweight framework utilizing compressed latent MSA representations from AlphaFold3 (Abramson et al., 2024) (AF3) and conditional Statistical Flow Matching (Cheng et al., 2025) (SFM) as a generative decoder to model the sequence distribution in an input MSA. Specifically, MSAFlow employs AF3's MSAModule as an encoder to produce pair representations of MSAs, which are then mean-pooled and used as conditioning for the SFM decoder that is trained to reconstruct the original set of sequences in the MSA (Figure 1). Unlike EVE (Frazer et al., 2021) which requires training a separate VAE for each MSA, MSAFlow learns a generalizable generative autoencoder over the space of MSAs (i.e., sets of sequences) with guaranteed permutation invariance. We further introduce a latent flow-matching model that generates MSA embeddings in a zero-shot manner from a single sequence's ESM embedding. By learning from homology-rich MSA representations, our latent flow-matching model can augment proteins with shallow or absent MSAs. Integrating these components, we provide a unified end-to-end framework capable of MSA representation, augmentation and family-based protein design.

We summarize our contributions as follows:

- **Novel architecture for modeling MSAs.** We propose MSAFlow, an generative autoencoding framework that operates on the sequence space. MSAFlow leverages compressed AF3 MSA embeddings to encode evolutionary information, paired with a conditional Statistical Flow-matching decoder that reconstructs MSA sequences while maintaining permutation invariance.

- We enabled **zero-shot generation of synthetic MSA** through a two-stage approach combining latent flow-matching over MSA embedding space and our MSAFlow decoder.

- We offer a **unified framework for MSA representation, augmentation, and family-based sequence design.** MSAFlow scales efficiently to large families, supports variable sequence lengths, and adapts flexibly to downstream design and analysis tasks—capabilities that prior models could not jointly achieve.

- **Empirical significance.** MSAFlow demonstrates competitive performance across multiple protein structure prediction and family-based protein design tasks, including zero-shot and few-shot MSA generation for orphan and low-homology proteins, and family-based enzyme design on EC classes with limited data. MSAFlow achieves these results despite being lightweight (130M parameters) and trained on smaller datasets, offering improved efficiency in terms of inference time and memory consumption (Table 7).

## 2 ADDITIONAL RELATED WORK

**Generative models for protein sequences**  Protein sequence generative modeling can be approached from both discrete and continuous perspectives. Discrete protein language models—such as autoregressive transformers and masked language models—treat amino acid as token, learning residue distributions through maximum likelihood estimation or masked denoising objectives. The ProGen series (Madani et al., 2020; Bhatnagar et al., 2025) and ESM (Lin et al., 2023b) represent notable examples that employ Transformer architectures (Vaswani et al., 2017) to model residue-residue dependencies across protein families. Recent research has also explored discrete diffusion frameworks, such as EvoDiff (Alamdari et al., 2023), which learns denoising processes in amino acid token space, generating sequences with desired structural or functional properties through sequential unmasking. Continuous methods, including flow-matching approaches like MultiFlow (Campbell et al., 2024) and FlowSeq (Ma et al., 2019), offer protein generation in continuous spaces. These continuous methods typically offer greater flexibility in conditional generation and interpolation but require decoding mechanisms to map continuous representations back to valid sequences. These language models have been applied to downstream tasks, including protein-binding peptide design (D-Flow (Wu et al., 2024), PepFlow (Li et al., 2024)), structure-based sequence design (LM-Design (Zheng et al., 2023), InstructPLM (Qiu et al., 2024), DRAKES (Wang et al., 2024)), and antibody engineering (Frey et al.). However, most existing approaches focus on single-sequence modeling and do not fully leverage evolutionary information contained in MSAs, which limits their capacity to capture residue co-variation and functional diversity essential for robust protein design.

**Latent diffusion for protein design.** Latent diffusion models were initially applied to protein structure generation, demonstrating advantages of continuous representations (Fu et al., 2024; Zhang et al., 2025; Xu et al., 2023; Yim et al.). Recently, these models have been used to model sequence–structure relationships through continuous embeddings. Latent spaces enable consistency across multiple protein modalities while maintaining compact representations. CHEAP (Lu et al., 2024) compresses protein embeddings via VAE or VQ techniques to create an efficient latent space. Building on this, PLAID (Lu et al.) applies latent diffusion over folding model embeddings for joint sequence–structure generation. Similarly, ProteinGenerator (Lisanza et al., 2024) performs diffusion in sequence space guided by RoseTTAFold (Baek et al., 2021) to enforce structural constraints. La-Proteina (Geffner et al., 2025a) further extends these capabilities using partially latent flow matching for scalable joint generation of sequences and all-atom structures. These advances have not yet been extended to MSAs, which capture evolutionary variation and residue-wise dependencies. MSAFlow addresses this gap by applying latent diffusion to the MSA domain.

## 3 METHOD

### 3.1 MSAFLOW: AN AUTO-ENCODING FRAMEWORK FOR MSAS

MSAs are mathematically represented as $\mathcal{S} = \{s_1, s_2, ..., s_M\}$ where each sequence $s_i \in \mathcal{A}^L$ consists of amino acids and gaps from alphabet $\mathcal{A}$, aligned to a reference sequence $s_{\text{ref}}$ of length $L$. Despite containing hundreds to thousands of sequences, we hypothesize that the functional and evolutionary

information within an MSA can be **compressed into a continuous latent representation** that captures the essential characteristics of the sequence distribution within that protein family.

This compression necessitates a permutation-invariant encoding method to avoid bias from sequence ordering. Formally, we seek an encoder $h_\phi : \mathcal{S} \to \mathbb{R}^d$ such that $h_\phi(\mathcal{S}) = h_\phi(\pi(\mathcal{S}))$ for any permutation $\pi$ of the sequences in $\mathcal{S}$. We leverage the AF3 MSAModule architecture, which provides a computationally efficient framework for embedding evolutionary information (Abramson et al., 2024). The AF3 MSAModule processes an MSA by

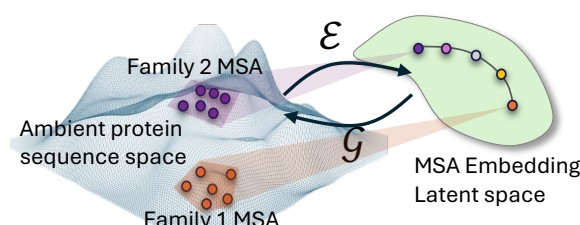

Figure 2: MSAFlow lifts autocoder to the space of sequence distributions within MSAs and families.

computing a position-wise outer product for each sequence $s_i$ with the reference sequence, resulting in pairwise representations $P_i \in \mathbb{R}^{L \times L \times h_{\text{pair}}}$. These representations are averaged across all sequences as $P_{\text{avg}} = \frac{1}{M} \sum_{i=1}^{M} P_i$. The averaged representation is then processed through multiple triangle self-attention blocks to produce a refined pair representation $P_{\text{refined}} \in \mathbb{R}^{L \times L \times H}$. We utilize Protenix (Team et al., 2025), a pretrained variant of AF3, to generate these embeddings for MSAs from the OpenFold dataset (Ahdritz et al., 2024). The resulting pair representation serves as our compressed MSA embedding $m = h_\phi(\mathcal{S}) \in \mathbb{R}^{L \times L \times H}$.

Viewed this way, MSAFlow realizes an autoencoding framework over sets: the encoder maps the finite set of sequences in an MSA to a latent embedding, while the decoder reconstructs the underlying family-level distribution of sequences conditioned on this latent representation. This perspective emphasizes that MSAFlow does not simply compress individual sequences, but rather learns a compact representation of the set as a distribution, enabling permutation-invariant and family-aware generative modeling.

### 3.1.1 STATISTICAL FLOW MATCHING FOR MSA SEQUENCE DECODING

We formulate MSA decoding as a conditional generation task over the sequences within a protein family. Given an MSA $\mathcal{S}$ and its embedding $m = h_\phi(\mathcal{S})$, the decoder reconstructs sequence distribution. Let $\tilde{\mathcal{S}} = \{s_1, \ldots, s_n\}$ be $n$ sequences drawn uniformly without replacement from $\mathcal{S}$. We model $p_\theta(\tilde{\mathcal{S}} \mid m) = \prod_{i=1}^{n} p_\theta(s_i \mid m)$, which is permutation-invariant by construction. The decoder $p_\theta(s \mid m)$ represents the probability of sampling a sequence $s$ compatible with $m$.

To instantiate $p_\theta(s \mid m)$ for discrete (categorical) sequences, we adopt Statistical Flow Matching (SFM) (Cheng et al., 2024), which learns a continuous Riemannian flow over the statistical manifold of categorical distributions equipped with Fisher-Rao metric. Concretely, each sequence in the MSA is treated as a sample of the target distribution. We operate in the probability simplex $\Delta^{|\mathcal{A}| \times L}$, where each position in the sequence is represented by a one-hot categorical distribution $\mu$ over amino acids.

Following SFM, we construct flow paths along geodesics on the positive orthant of the unit sphere by applying the mapping: $\pi : x = \pi(\mu) = \sqrt{\mu}$. SFM demonstrated that such a mapping to the unit sphere preserves the metric, which coincides with the canonical spherical geometry. Therefore, we can operate on the unit sphere with the standard spherical geometry. Mathematically, given a sequence $s_i$ from the MSA and its corresponding categorical representation $x_1 = \pi(\mu_1)$ (e.g., one-hot encoding) and the noise representation $x_0 = \pi(\mu_0)$, the time-dependent interpolation follows:

$$x_t = \exp_{x_0}(t \cdot \log_{x_0}(x_1)) \tag{1}$$

where $\exp$ and $\log$ are the spherical exponential and logarithm maps on the manifold, respectively, and can be calculated in closed form as

$$\exp_x(u) = x \cos \|u\|_2 + \frac{u}{\|u\|_2} \sin \|u\|_2, \quad \log_x(y) = \frac{\arccos(\langle x, y \rangle)}{\sqrt{1 - \langle x, y \rangle^2}}(y - x - \langle x, y - x \rangle x), \tag{2}$$

After transforming back to the simplex with $\mu_t = \pi^{-1}(x_t)$, the interpolation in Equation 1 traces the geodesic between $\mu_0$ and $\mu_1$ with respect to the Fisher information metric, ensuring we follow the

shortest path on the statistical manifold. The corresponding vector field for this mapped geodesic flow is given by $u_t(x_t|x_0, x_1) = \log_{x_t}(x_1)/(1-t)$. Instead of an unconditional model, our MSAFlow decoder employs a conditional parameterization where $v_\theta(x_t|m, t)$ is trained to approximate the vector field conditioning on the MSA embedding $m = h_\phi(\mathcal{S})$:

$$\mathcal{L}_{\text{SFM}}(\theta) = \mathbb{E}_{t\sim\mathcal{U}[0,1], s_i\sim\mathcal{S}, \mu_0\sim\pi_*p_0, \mu_1\sim\pi_*\delta(s_i)}\left[\|v_\theta(x_t|m, t) - u_t(x_t|x_0, x_1)\|^2\right] \quad (3)$$

where $\pi_*$ denotes the pushforward operation of applying the mapping $\pi$, $x_t$ is obtained via the geodesic interpolation, and $\delta(s_i)$ represents the categorical distribution corresponding to sequence $s_i$ (typically a one-hot encoding) in an MSA. During sampling, we first follow the learned marginal vector field on the sphere to obtain $x_1$, then discrete generations of MSAs can be sampled from the categorical distribution $\mu_1 = \pi^{-1}(x_1)$.

### 3.1.2 MODEL ARCHITECTURE AND IMPLEMENTATION

We implement the vector field model $v_\theta$ using a modified conditional Diffusion Transformer (DiT) (Peebles & Xie, 2023) architecture. Since the output of the AF3 MSAModule is the pair representation of dimension $L \times L \times H$, we first compress it along the second dimension through mean pooling to obtain a sequence-level representation of dimension $L \times H$:

$$m_{\text{seq}} = \frac{1}{L}\sum_{j=1}^{L} m_{:,j,:} \in \mathbb{R}^{L\times H} \quad (4)$$

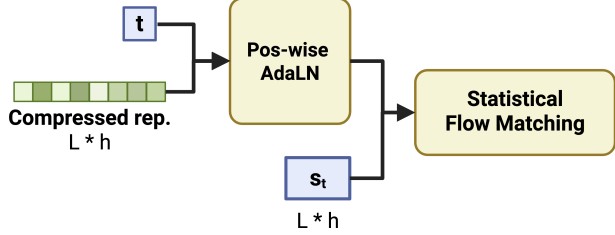

Figure 3: DiT architecture for MSAFlow decoder.

This compressed representation serves as conditional information for the DiT model, which consists of 12 transformer blocks with a hidden dimension of 768, totaling approximately 130M parameters. The architecture incorporates sinusoidal time embeddings for the diffusion timestep $t$, token embeddings for each amino acid position, conditional embeddings from the compressed MSA representation, and multi-headed self-attention blocks with adaptive layer normalization. Notably, the MSA embedding conditioning is applied per-residue through a **position-wise AdaLN**, which introduces a novel mechanism for residue-level control. Unlike global conditioning schemes that broadcast the same modulation across all tokens, this design injects fine-grained, position-specific information into each layer normalization step, allowing for more precise alignment between evolutionary context and sequence generation. This innovation enhances the expressivity of the conditioning pathway and represents a new approach for leveraging MSAs in diffusion-based protein design. At inference time, we sample sequences by starting with random noise $x_1 \sim \text{Uniform}(\mathcal{A})$ and iteratively applying:

$$x_{t-\Delta t} = x_t - v_\theta(x_t|m, t) \cdot \Delta t \quad (5)$$

for timesteps $t = 1, 1-\Delta t, 1-2\Delta t, ..., 0$, where $\Delta t$ is a small step size (typically 0.01). At $t = 0$, we obtain the final sequence by taking the argmax over the amino acid probabilities at each position.

### 3.2 CONDITIONAL LATENT FLOW MATCHING FOR ZERO-SHOT MSA EMBEDDING GENERATION

While our decoder model generates sequences from MSA embeddings, we also develop a complementary approach to generate synthetic MSA embeddings themselves. This enables us to create artificial MSAs for proteins with limited evolutionary data. Let $z_1 = h_\phi(\mathcal{S}) \in \mathbb{R}^{L\times H}$ be the compressed MSA embedding for a reference sequence $s_{\text{ref}}$, and let $e = g_\psi(s_{\text{ref}}) \in \mathbb{R}^{d_e}$ be its ESM embedding. We aim to learn a conditional generative model $p_\theta(z_1|e)$ that can produce plausible MSA embeddings given only the reference sequence embedding.

**Latent Flow Matching:** We train a *conditional rectified flow* that maps a standard Gaussian $z_0 \sim \mathcal{N}(0, I)$ on the distribution of MSA embeddings $p(z \mid e)$ conditioned on the ESM embedding $e$ (Lin et al., 2023b). We use a straight-line path $z_t = (1-t)z_1 + t z_0$ from target $z_1$ (the ground-truth MSA embedding) to noise $z_0$, whose reference velocity is the constant field $u_t^\star(z_t; z_0, z_1) = z_0 - z_1$. A time-dependent, conditional velocity $v_\theta(z_t, e, t)$ is learned by least-squares flow matching:

$$\mathcal{L}_{\text{RFM}} = \mathbb{E}_{t\sim\mathcal{U}[0,1], z_0\sim\mathcal{N}(0,I), z_1}\left\| v_\theta(z_t, e, t) - (z_0 - z_1)\right\|_2^2, \quad (6)$$

which provides a simple, stable objective without explicit score estimation.

**Generative Sampling Process:** At inference, we draw $z_0 \sim \mathcal{N}(0, I)$ and integrate the learned conditional velocity backward from $t=1$ to $t=0$ with an explicit Euler solver. By default we use the deterministic probability-flow ODE ($T=0$); optionally, we add isotropic noise with temperature $T \in [0, 1]$ to trade fidelity for diversity:

$$z_{t-\Delta t} = z_t - v_\theta(z_t, e, t)\, \Delta t \; + \; T \sqrt{\Delta t}\, \varepsilon, \qquad \varepsilon \sim \mathcal{N}(0, I). \tag{7}$$

Empirically, smaller $T$ (e.g., $T=0.5$) improves alignment to $e$, while larger $T$ increases sample diversity. Full SDE variants and discretization details are provided in Appendix 6.7.

### 3.3 END-TO-END UNIFIED PIPELINE FOR MSA REPRESENTATION, AUGMENTATION AND FAMILY-BASED SEQUENCE DESIGN

Our complete framework enables three complementary paths for MSA generation (as shown in Figure 1), each tailored to specific protein scenarios:

**MSA Compression and Reconstruction:** For deep MSAs with abundant evolutionary information, we first compress the multidimensional sequence information through the AF3 MSAModule into a compact latent representation. This compressed embedding effectively captures the evolutionary and functional signals present in the original MSA. We then use our SFM decoder to selectively reconstruct sequences, maintaining the key evolutionary characteristics while reducing redundancy.

**Zero-shot MSA Augmentation:** For orphan or de novo proteins with limited evolutionary data, we first generate the ESM embedding of the single available sequence. Our latent diffusion model then transforms this single-sequence representation into a synthetic MSA embedding that emulates the evolutionary diversity typically found in natural protein families. Finally, we decode multiple diverse sequences from this embedding using our SFM decoder, effectively bootstrapping evolutionary information where none previously existed.

**Family-based Sequence Design:** To perform family-based protein design, we first align all sequences belonging to the family (e.g., enzyme class) for a given query. These sequences are compressed into a latent representation using our AF3-based MSA encoder. Our SFM decoder then generates new sequences conditioned on this latent embedding, effectively producing new sequence designs that share a similar distribution to the given family. Because the generated sequences may include gaps, we can support both variable-length and fixed-length designs: gaps can be ignored when constructing the final sequence, enabling flexible design strategies.

This approach combines both MSA compression and generation capabilities in a unified framework. For data-rich scenarios, our method enables efficient information extraction from deep MSAs while preserving their evolutionary signals. For data-limited proteins, it allows the creation of synthetic alignments that capture potential evolutionary diversity. The integration of these complementary pathways addresses a fundamental limitation in protein analysis by extending evolutionary context to proteins that previously lacked sufficient homologous sequences, potentially improving downstream structure prediction, functional annotation tasks, and family-based design ability.

## 4 EXPERIMENTS

### 4.1 BENCHMARKING MSA AUTOENCODING

We evaluate the reconstruction capability of our model on 50 proteins released by CAMEO on May 10, 2025, where the ground truth MSA is generated using the same procedure as described in (Team et al., 2025). We took rigorous measures to avoid data leakage (maximum sequence identity from training set of 0.72, average 0.55) and ensured clear temporal separation between training and evaluation sets as described in Appendix 6.1. We compute the embedding for each MSA using the AF3 MSAModule and generate 32 sequences for each latent MSA representation. The shallow MSAs generated by our model achieve structure prediction metrics approaching those of deep, ground-truth MSAs in terms of pLDDT (89.0 vs. 91.6) and TM-scores (0.86 vs. 0.89) while consuming 6.5% of the storage required to represent a deep MSA. This compression ratio corresponds to an average sequence length of 365 and more than 7,000 alignments from the CAMEO dataset. We perform

conditional generation given an embedding of 16-bit floats with an average size of 365×128 from the CAMEO dataset.

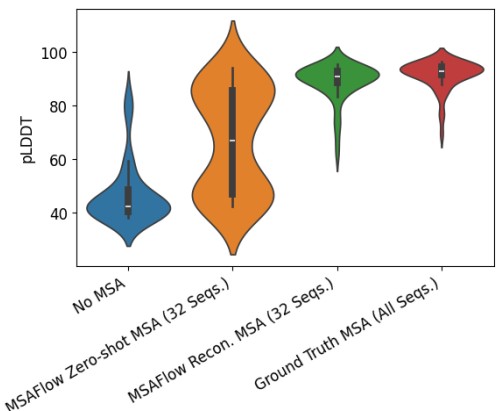 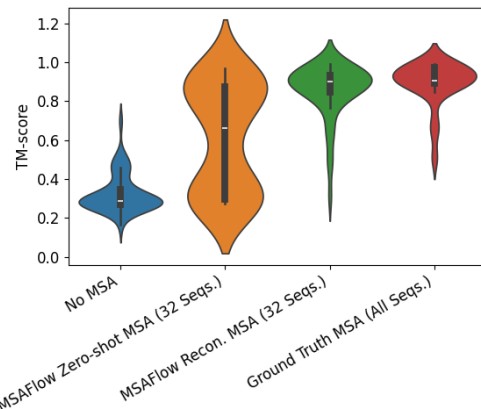

Figure 4: pLDDT and TM-scores for AF3 predictions of proteins from CAMEO with no MSA, MSAs generated through the MSAFlow-based zero-shot augmentation method, the MSAFlow-based reconstructed MSA (32 sequences), and the ground truth deep MSA (approximately 7k sequences).

When evaluating synthetic MSA embeddings generated via our latent diffusion model, we find that our decoder reconstructs meaningful signals from the generated MSA latents, achieving higher quality than predictions without MSAs, though structure prediction accuracy remains below that obtained using ground truth embeddings. Our model compresses evolutionary information encoded in thousands of aligned sequences into a single, fixed-size latent tensor that can be decoded into sequences that remain evolutionarily related to the query, as further evidenced in Table 9. This compression preserves most of the functional signal relevant for folding accuracy. Moreover, synthetic MSAs consistently improve structure prediction over the no-MSA baseline: across the CAMEO benchmark, zeroshot-generated MSAs improve pLDDT in 97.96% of cases and improve TM-score in 89.80% of cases, indicating that failure cases are rare. For completeness, we provide in the appendix the PDB IDs corresponding to the small fraction of proteins where no improvement is observed.

We further evaluate the intrinsic quality of generated MSAs by comparing their residue-level entropy statistics to ground truth alignments. Following the evaluation setup in Zhang et al. (2024b), we generate 1000 sequences per MSA for our CAMEO test set and compute per-position entropies. MSAFlow's alignments mirror ground-truth entropy profiles, with generated sequences centered on the true distribution (average entropy difference of 0.076 vs. 0.136 for ProfileBFN). Residue-level conservation patterns are preserved with high fidelity, as reflected by lower variance (0.294 vs. 0.724), demonstrating that MSAFlow achieves alignment quality closer to ground truth statistics. To more completely characterize distributional similarity beyond first- and second-order statistics, we additionally compute Wasserstein distance and Maximum Mean Discrepancy (MMD), two well-established divergence metrics for comparing distributions. Across both metrics, MSAFlow exhibits substantially lower divergence from GT MSAs than ProfileBFN, reinforcing that our reconstructed MSAs more faithfully preserve the underlying evolutionary signal.

Table 1: Comparison of entropy and distributional statistics between generated MSAs and ground truth (GT). MSAFlow more accurately recapitulates GT distributions, with lower entropy deviations, Wasserstein distance, and MMD.

|  | MSAFlow | ProfileBFN | GT |
|---|---|---|---|
| Average entropy | $2.755 \pm 0.294$ | $2.838 \pm 0.724$ | $2.68 \pm 0.589$ |
| Average entropy difference from GT | **0.076** | 0.136 | – |
| Average Wasserstein distance from GT | **0.344** | 0.470 | – |
| Average MMD from GT | **0.541** | 0.875 | – |

## 4.2 AUGMENTING SHALLOW AND SINGLE-SEQUENCE MSAS

We evaluate our model on a dataset of sequences with limited evolutionary information derived from MSAGPT (Chen et al., 2024), which includes 200 proteins from CAMEO (Haas et al., 2018), CASP14, CASP15, and PDB (Berman et al., 2000) with either few or no sequences in their MSA (few-shot and zero-shot cases, respectively). For the zero-shot case, we embed the query sequence with ESM and use it as conditioning for our latent diffusion model, which generates a synthetic MSA embedding for the reference sequence. We generate embeddings using 10 different seeds and employ low-temperature sampling during the SDE forward pass for higher-fidelity reconstructions, as detailed in (Geffner et al., 2025b). We then decode 32 sequences from each of the 10 synthetic MSA embeddings and report the best pLDDT and TM-scores. Our model achieves improved performance compared to prior MSA augmentation methods when evaluated using AF3.

Table 2: The accuracy of MSAFlow-generated multiple sequence alignments compared to other state-of-the-art methods, as evaluated by AlphaFold3 protein structure prediction performance on a naturally scarce MSA dataset curated from CAMEO, PDB, and CASP.

| | AF3 pLDDT | | TM-score | |
|---|---|---|---|---|
| | *Zero-shot* | *Few-shot* | *Zero-shot* | *Few-shot* |
| No/Shallow MSA | 73.1 | **70.8** | 0.55 | 0.58 |
| EvoDiff (650M) | 67.7 | 67.5 | 0.49 | 0.55 |
| MSAGPT (3B) | 71.6 | 70.3 | 0.53 | 0.58 |
| ESMFold | - | - | 0.58 | - |
| MSAFlow (Ours,130M) | **75.2** | 70.4 | **0.62** | **0.60** |

For the few-shot augmentation case, we use our latent flow matching model to generate synthetic embeddings for each sequence over 5 different seeds, and decode 32 sequences from each MSA embedding. We then decode 64 sequences from the ground-truth shallow MSA embedding and extract the 16 most diverse sequences across all generations, following Chen et al. (2024). We concatenate our generated sequences with the original shallow MSA and observe improvements in structure prediction accuracy for these cases. We detail ablations motivating this reconstruction and augmentation scheme in Appendix 6.5 and 6.6.

## 4.3 CASE STUDIES ON *de novo* AND INTRINSICALLY DISORDERED PROTEINS

We demonstrate that MSAFlow improves structure prediction for challenging proteins by generating synthetic MSAs. We focus on three cases from a sparse MSA dataset:

- **8B4K**: the N-terminal domain of Rfa1 complexed with a phosphorylated Ddc2 peptide—only 133 residues, with scarce evolutionary relatives.
- **8G8I**: a Rosetta-designed four-helix bundle with rigid backbone constraints, extraordinary thermal stability ($T_m > 90°C$), and NMR-validated topology (backbone RMSD = 1.11 Å).
- **8OKH**: the crystal structure of *Bdellovibrio bacteriovorus* Bd1399.

MSAFlow's synthetic MSAs outperform both MSA-free predictions and those using MSAGPT. This demonstrates MSAFlow's capability in addressing two challenging scenarios: (i) limited sequence homology and (ii) intrinsically flexible or disordered regions. By generating MSAs in latent space, our method provides evolutionary signals that modern folding models require for these targets. We provide additional case studies in Appendix 6.2.

## 4.4 FAMILY-BASED PROTEIN DESIGN

To better demonstrate the strength of MSAFlow on few-shot generation and generalization to other downstream applications than AF3 prediction, we now provide new results on family-based enzyme design. **Our experiments demonstrate clear and significant advantages of MSAFlow, particularly for EC classes with limited sequences.** Following ProfileBFN (Gong et al., 2025), we generate sequences in a single shot using our model, for enzymes with less than 20 sequences in their corresponding EC class, using the sequences from the EC class as an MSA. We then use CLEAN (Yu

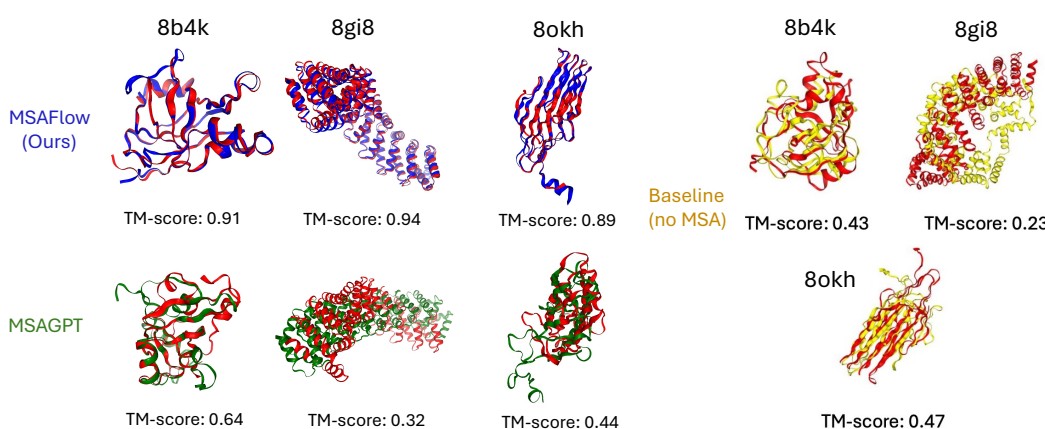

Figure 5: Visualization of improved structure prediction for zero-shot augmentation on de novo and disordered proteins with MSAFlow-decoded synthetic MSAs, as compared to MSAs generated with MSAGPT. Blue represents predictions with an MSAFlow-generated MSA and green represents predictions with an MSAGPT-generated MSA. Red indicates the ground truth structure, and yellow indicates the prediction obtained without using any MSA.

et al., 2023) to determine their EC number, and compute the accuracy (i.e. how many generated designs match the ground truth EC number) and the uniqueness across all generated designs. We report the accuracy × uniqueness score as done by ProfileBFN, the current SOTA for this task. **MSAFlow exhibits SOTA performance on family-based enzyme design in both fixed and variable length settings.** Notably, ProfileBFN is confined to fixed-length generation, whereas MSAFlow learns a meaningful homology distribution that guides the placement of gaps, which effectively enables variable-length design with unprecedented success rate.

Table 3: Performance comparison of MSAFlow with baseline methods on family-based enzyme design task across different EC classes.

|  |  | Q15I65 | Q15BH7 | P13280 | P57298 |
|---|---|---|---|---|---|
| **MSA Depth** |  | 15 | 12 | 13 | 15 |
| **# of Generated Sequences** |  | 1000 | 100 | 100 | 100 |
| | EvoDiff | 1.39% (Gong et al., 2025) | 0% | 80% | 5% |
| **Acc. × Uniqueness (Fixed Length)** | ProfileBFN | 42.67% (Gong et al., 2025) | **89%** | **100%** | 82% |
| | MSAFlow | **83.10%** | 84% | **100%** | **95%** |
| | EvoDiff | 0% | 0% | 0% | 0% |
| **Acc. × Uniqueness (Variable Length)** | MSAGPT | 15.11% | 35.59% | 37.5% | 24.98% |
| | MSAFlow | **51%** | **92%** | **92%** | **84%** |

To further validate that MSAFlow generates evolutionarily meaningful and non-degenerate variants, we additionally evaluate Diversity and Novelty metrics across three enzyme families (Table 4). Diversity measures average pairwise sequence identity among generated sequences (lower indicates greater diversity), while Novelty measures dissimilarity from the natural sequence (higher indicates more distinct yet plausible variants). As shown below, compared to MSAGPT and ProfileBFN, MSAFlow achieves the strongest balance of evolutionary diversity and novelty across all enzyme classes. Although the diversity of novelty of EvoDiff is also high, its accuracy is close to 0% in Table 3, indicating that many of the generated MSAs are irrelevant random sequences that do not capture the evolutionary information well. In contrast, MSAFlow consistently generates high-quality MSAs while maintaining broad diversity and novelty.

## 5 CONCLUSION

MSAFlow integrates statistical flow matching with latent space optimization to enable bidirectional manipulation of multiple sequence alignments. By combining AlphaFold3-inspired permutation-equivariant embeddings with diffusion-based generation, it uniquely achieves both evolutionary

Table 4: Diversity (lower = more diverse) and Novelty (higher = more distinct from natural sequence) of generated sequences across three enzymes. MSAFlow achieves one of the strongest combinations of diversity and novelty. Although EvoDiff also achieves good diversity, its close to 0% accuracy limits the practical application. Best results are in **bold** and the second best are underlined.

| Metric | Model | P13280 | P57298 | Q15BH7 | Q15165 |
|---|---|---|---|---|---|
| Diversity↓ | MSAFlow (ours) | 0.100 | 0.150 | 0.117 | **0.434** |
| | EvoDiff | **0.062** | **0.064** | **0.064** | 0.788 |
| | MSAGPT | 0.834 | 0.896 | 0.622 | 0.838 |
| | ProfileBFN | 0.392 | 0.271 | 0.360 | 0.594 |
| Novelty↑ | MSAFlow (ours) | 0.834 | 0.901 | 0.781 | 0.420 |
| | EvoDiff | **0.898** | 0.895 | **0.897** | **0.922** |
| | MSAGPT | 0.184 | 0.894 | 0.228 | 0.099 |
| | ProfileBFN | 0.601 | **0.902** | 0.644 | 0.288 |

signal compression and biologically plausible augmentation of sparse alignments. Comprehensive benchmarking across critical applications—latent space reconstruction fidelity, shallow MSA augmentation, synthetic alignment generation, and enzyme design—demonstrates MSAFlow's superiority, achieving state-of-the-art performance with only 130M parameters. MSAFlow's ability to generate evolutionarily coherent sequence ensembles creates new opportunities for designing orphan proteins and tackling de novo structure prediction challenges. Importantly, our framework also enables family-based design, where latent representations distilled from enzyme or protein families can guide the generation of sequences that remain faithful to family-level constraints while still exploring novel sequence diversity. Overall, MSAFlow advances both computational efficiency and conceptual modeling of protein sequence spaces through flow-based generation, paving the way for conditional protein engineering, resource-efficient applications, and family-level design of functional proteins.

## ETHICS AND REPRODUCIBILITY STATEMENT

We have taken several steps to ensure reproducibility of our findings. The full model description, including encoder, decoder, and flow-matching components, is detailed in Section 3. Hyperparameters, training/test splits, and dataset sources are provided in Section 6.1. Ablation studies (Section 6.6, Table 9) clarify the contributions of different components, and additional case studies (Section 6.2, Table 4) demonstrate robustness across diverse proteins. Experimental comparisons with baselines are presented in Section 4 (Tables 1–3). To further facilitate reproducibility, we plan to release the anonymized source code and trained models in the supplementary material soon.

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

# 6 ADDITIONAL RESULTS

## 6.1 DETAILS ON MSAFLOW TRAIN/TEST SPLIT

The maximum sequence identity for sequences in our CAMEO reconstruction dataset to our training set is 0.72, when run at 80% coverage against the consensus sequence for each MSA in the training set (with the average maximum identity across all sequences in the test set being 0.55). This is an even stricter threshold than MSAGPT (which uses 90% coverage instead). Furthermore, the MSAs we used for training come from the OpenProteinSet, which consists of sequences searched from Uniclust30 v2018-8. The cutoff for AlphaFold3 training data is September of 2021, and the cutoff for ESM2 training data is February of 2020. The CAMEO structures we used for reconstruction evaluation, however, were all deposited in May of 2025. This rigorous separation ensures the novelty of our test set. This is in line with ProfileBFN, which trains on the same corpus as ESM2, while evaluating their model on CAMEO structures deposited in May of 2024. For the zero-shot/few-shot augmentation task, we use the same test set as MSAGPT, which is also trained on the OpenProteinSet. The authors ensure minimal data leakage between the train and test set during their experiments, which implies the same for MSAFlow.

## 6.2 ADDITIONAL CASE STUDIES

To further validate the robustness of MSAFlow's zero-shot predictions, we provide more cases for comparison. From the table 5, we can observe that MSAFlow achieves improvement on cases with different structural patterns as well as different families. We also provide the ground-truth zero-shot prediction folding accuracy for the case studies in Figure **??**.

| PDB ID | Length | Description | GT | MSAGPT | MSAFlow |
|--------|--------|-------------|------|--------|---------|
| 6NW8_A | 27 | Scorpion venom toxin | 0.39 | 0.40 | **0.53** |
| 6WKK_X | 280 | Phage capsid | 0.28 | 0.27 | **0.55** |
| 7EQB_B | 80 | Central spindle assembly | 0.65 | 0.58 | **0.71** |
| 7QRR_L | 153 | Noumeavirus | 0.31 | 0.61 | **0.83** |
| 7ZOL_A | 151 | Cas 7-11 regulator | 0.33 | 0.34 | **0.67** |

Table 5: Performance comparison of MSAFlow with baseline methods on clinically relevant proteins showing TM-Score improvements across different structural patterns and protein families.

| PDB ID | GT | MSAGPT | MSAFlow |
|--------|------|--------|---------|
| 8OKH | 0.47 | 0.64 | **0.89** |
| 8GI8 | 0.23 | 0.32 | **0.94** |
| 8B4K | 0.43 | 0.44 | **0.91** |

Table 6: Performance comparison of MSAFlow with ground-truth for the case study in Figure **??**.

## 6.3 INFERENCE SPEED AND MEMORY COST

In order to demonstrate that MSAFlow exhibits notable improvements in sampling efficiency compared to other MSA-based generative models, We benchmark MSAFlow against existing tools, attempting to generate 100 sequences conditioned on an existing MSA with 6 sequences on an NVIDIA A40 GPU, and observe the following:

We find that MSAFlow has better sampling efficiency, both in terms of speed and memory. We can attribute this to the fact that our model only has to deal with L×H embedding of the MSA, rather than carry the quadratic cost of representing an MSA in the ambient space. The result shows that MSAFlow has the potential to be a highly light-weight and accurate MSA designer.

Moreover, our pipeline utilizes outputs from tools like MMseqs and HMMER for Multiple Sequence Alignment (MSA) reconstruction. A key advantage of this approach is its ability to generate high-quality MSAs even when these standard homology search methods fail to find sufficient homologous

|            | Latency Per Sequence | Memory Consumption |
|------------|----------------------|--------------------|
| MSAFlow    | **1.02s**            | **5.8 GiB**        |
| ProfileBFN | 8.49s                | 7.7 GiB            |
| MSAGPT     | 62.46s               | 41.6 GiB           |
| EvoDiff    | 478.24s              | 4.0 GiB            |

Table 7: Sampling efficiency comparison of MSAFlow with baseline methods showing latency per sequence and memory consumption on NVIDIA A40 GPU for generating 100 sequences conditioned on an MSA with 6 sequences.

information. To provide a quantitative comparison of computational cost, we evaluated our MSAFlow model against HMMER and MMseqs2 for generating an MSA from a single query sequence (PDB 9BCZ_A from CAMEO, 644 amino acids). The empirical results are detailed below.

| Method             | Wall Clock Time (s) |
|--------------------|---------------------|
| MSAFlow (100 seqs) | **153.93**          |
| HMMER              | 310.92              |
| MMseqs2            | 497.73              |

Table 8: Computational cost comparison for generating MSA from query sequence alone (PDB 9BCZ_A from CAMEO, 644 AA) showing wall clock time in seconds.

These results show that MSAFlow achieves over $2\times$ speedups compared to HMMER and MMseqs2, while still providing the ability to operate in settings where homology search fails. This confirms that MSAFlow not only addresses the coverage gap but also offers computational efficiency advantages over traditional methods.

### 6.4 ABLATION STUDY OF RECONSTRUCTION SEQUENCES

We address using the additional ablation study on the reconstuction task with 2, 4, 8, 16, and 32 decoded MSA sequences, as well as the comparison with natural-MSA depth on 3 samples from the CAMEO reconstruction test set.

When we keep 2-4 sequences, the MSAFlow reconstructions beat the random ground-truth subsample. As we generate more sequences, the designed MSAs generally match that of the ground-truth samples (AlphaFold3 searched MSA), indicating that MSAFlow accurately captures structure patterns of protein families.

|                              | PDB ID | 2        | 4        | 8    | 16       | 32   |
|------------------------------|--------|----------|----------|------|----------|------|
| **Ground Truth Random Sample** | 9EJY   | 0.59     | 0.55     | 0.85 | 0.80     | 0.86 |
|                              | 9BIX   | 0.19     | 0.32     | 0.35 | 0.32     | 0.49 |
|                              | 9CVV   | 0.35     | 0.31     | 0.93 | 0.97     | 0.98 |
| **MSAFlow Reconstruction**   | 9EJY   | **0.61** | **0.61** | 0.84 | **0.83** | 0.84 |
|                              | 9BIX   | **0.28** | 0.22     | 0.20 | 0.30     | 0.26 |
|                              | 9CVV   | **0.43** | **0.62** | 0.87 | 0.87     | 0.97 |

Table 9: Ablation study comparing MSAFlow reconstruction performance against ground truth random samples across different sequence counts on CAMEO reconstruction test set. Values represent performance metrics for MSA reconstruction quality. Numbers in the first row denotes the amounts of decoding MSA sequences.

## 6.5 ABLATION STUDY ON SYNTHETIC AND RECONSTRUCTED MSAs

The reconstruction pathway preserves the authentic signal from a limited, shallow MSA, while the latentflow pathway generates evolutionary diversity generalized from other MSA-rich proteins. These two tracks provide complementary signals that make the few-shot augmentation stronger. To provide evidence for this, we detail the separate contributions of each track below:

| Few-shot task | TM Score | Avg Per-position Entropy |
|---|---|---|
| **Syn-16** | 0.54 | 2.23 |
| **Rec-16** | 0.52 | 1.33 |
| **Syn+Rec-32** | **0.57** | 2.69 |
| **Syn+Rec+GT** | **0.60** | 2.58 |
| **MSAGPT+GT** | 0.58 | 1.33 |
| **GT** | 0.58 | 2.16 |

Table 10: Ablation study showing the complementary contributions of synthetic and reconstructed MSA pathways in few-shot tasks, demonstrating improved TM scores and entropy characteristics. **Syn** represents Synthetic MSAs; **Rec** represents Reconstructed MSAs. The number denotes amount of MSA sequences.

As shown in the table, the reconstruction path focuses on preserving crucial motif information within the limited observed sequences, which is reflected in the lower entropy signals in the shallow MSA. In contrast, the latentflow path generates synthetic MSAs that provide evolution-consistent diversity, resulting in higher entropy.

The combination of both tracks leads to an improvement in TM score and an increase in entropy. This observation confirms that the two tracks offer complementary signals, which synergistically improve quality. Finally, by augmenting the shallow ground truth MSA with the combined generation output, we improve prediction accuracy and achieve a better TM score than the MSAGPT baseline, which is what we report in Table 1. As can be seen, MSAFlow is the only method to achieve a better TM score than the ground truth, with an entropy value closest to it.

## 6.6 ABLATION STUDY ON ESM EMBEDDINGS

To clarify the individual contributions of the ESM embeddings and our proposed Statistical Flow-matching decoding mechanism, we perform an ablation study on the zero-shot augmentation track of MSAFlow. Specifically, we compare:

• A simple feature regression task that learns MSA embeddings from ESM2 features

• Replacing ESM2 embeddings with one-hot encodings of the query sequence

• Full ESM2 embeddings with our latent statistical flow-matching decoder

| Method | TM Score |
|---|---|
| MSAGPT (3B) | 0.53 |
| MSAFlow Latent w/ ESM2 regression (128M) | 0.54 |
| MSAFlow Latent w/ one-hot (130M) | 0.55 |
| MSAFlow Latent w/ ESM2 (130M) | **0.62** |

Table 11: Ablation study comparing the contribution of ESM embeddings versus one-hot sequence encoding in MSAFlow's zero-shot MSA augmentation performance.

The results demonstrate that the efficiency of our method. Moreover, ESM2 encoding provides more useful signals to address the evolutionary information.

## 6.7 GENERATIVE SAMPLING PROCESS

To sample a synthetic MSA embedding, we convert the ODE flow into an SDE following Geffner et al. (2025a), and integrate the reverse-time stochastic differential equation:

$$dz_t = \left(v_t^\theta - \frac{1}{2}g_t^2 \cdot s_t^\theta(x_t)\right)dt + T \cdot g_t \cdot d\bar{W}_t, \quad t \in [0,1] \tag{8}$$

where $f_t = -\frac{z_t}{1-t}$ is the drift term of the forward rectified flow, $g_t = \sqrt{\frac{2t}{1-t}}$ is the diffusion coefficient, $s_t^\theta = \nabla_{z_t} \log p_\theta(z_t|e,t)$ is the score function that can be converted from our predicted $v^\theta$, $T \in [0,1]$ is a temperature parameter, and $d\bar{W}_t$ is the standard Wiener process running backward in time.

We implement the sampling using the Euler-Maruyama discretization with steps of size $\Delta t$:

$$z_{t-\Delta t} = z_t - \left(v_t^\theta - \frac{1}{2}g_t^2 \cdot s^\theta(z_t,e,t)\right)\Delta t - T \cdot g_t\sqrt{\Delta t} \cdot \varepsilon, \quad \varepsilon \sim \mathcal{N}(0,I) \tag{9}$$

where $v_\theta(z_t,e,t)$ is the time-dependent vector field predicted by the DiT. The temperature parameter $T$ controls the stochasticity of the generation: $T = 1$ reproduces the exact generative SDE used during training, while $T \to 0$ suppresses the noise, approaching the deterministic probability-flow ODE.

## 6.8 IMPLEMENTATION DETAILS

### 6.8.1 DATASET PREPARATION

We use the OpenFold dataset Ahdritz et al. (2024), which consists of 16M MSAs in total. To filter high-quality MSAs, we only use alignments which have at least 10 sequences where at most 10% of the sequence consists of gaps, following Chen et al. (2024). This results in a dataset of 4M MSAs. We then generate MSA embeddings for each MSA with Protenix Team et al. (2025). Specifically, we truncate the inference framework to halt after the MSAModule step and dump the corresponding pair representation of the query sequence. This results in an embedding of shape $(L \times 128)$, where $L$ is the length of the query sequence. We use the default parameters that are used for structure prediction for this task.

### 6.8.2 TRAINING

We train a 130M parameter latent flow matching model and a 129M parameter conditional statistical flow matching decoder model. Both the encoder and decoder model are in congruence with the medium-size architecture of a diffusion transformer detailed in Peebles & Xie (2023). We detail their architectures in Table 2. For the encoder, our objective is to reconstruct the MSA embedding conditioned on the query sequence. For enhanced conditioning signal, we use the ESM2 650M Lin et al. (2023a) to generate an embedding of the query sequence, resulting in a tensor of shape $(L \times 1280)$. We train our encoder for 15 epochs on four H200 GPUs using the Adam optimizer with a learning rate warmup of 3000 steps, learning rate of 2.6e-4, and a weight decay of 0.1. We use a batch size of 32,768 maximum total sequence length for all sequences in the batch.

Our decoder model is conditioned on the MSA embedding and learns to reconstruct one-hot encoded sequences from the original MSA. Our vocabulary includes the 20 standard amino acids, as well as the gap token and the unknown amino acid token (X). We select only 32 sequences to reconstruct per MSA, where each sequence is weighted to compensate for the data bias present in MSAs. Specifically, each sequence's weight $w_i$ is computed as follows:

$$w_i = \left(1 + \sum_{j \neq i} \mathbf{1}\left\{d_{\text{hamming}}(x_i, x_j) < 0.2\right\}\right)^{-1}$$

This reweighting scheme reduces the influence of clusters of highly similar sequences, and is used by Rao et al. (2021). We train our decoder for 7 epochs on four H200 GPUs with a learning rate warmup of 5000 steps, learning rate of 1e-5, and a weight decay of 0.1. We use a batch size of 2,560 maximum total query sequence length for all MSAs in the batch, as each MSA has 32 sequences for reconstruction as well.

### 6.8.3 EVALUATION

For the folding task, we evaluate our MSAs by folding each sequence with the same seed, with 200 diffusion steps, 10 cycles, and 1 diffusion sampling trajectory. These are the default parameters provided by Protenix. For evaluating ESMFold Lin et al. (2023a), we use the standard implementation provided by the Transformers library.

Table 12: Architecture parameters of MSAFlow

| Latent FM Encoder | SFM Decoder |
|---|---|
| Input: $FC(in = 128, out = 768)$ | Input: $FC(in = 22, out = 768)$ |
| Conditioning: $FC(in\ dim. = 1280, out\ dim. = 768)$ | Conditioning: $FC(in = 128, out = 768)$ |
| $12\times DiTAdaLn(in = 768, heads = 12, cond. = 768)$ | $12\times DiTAdaLn(in = 768, heads = 12, cond. = 768)$ |
| Output: $FC(in = 768, out = 128)$ | Output: $FC(in = 768, out = 22)$ |

## 6.9 ABLATION STUDY ON DECODER ARCHITECTURE AND CONDITIONING MECHANISM

To further evaluate the flexibility of our framework and isolate the contributions of individual architectural components, we introduce new ablation studies spanning (i) alternative discrete diffusion formulations and (ii) alternative conditioning mechanisms.

### 6.9.1 COMPATIBILITY WITH MDLM AND SUPERIORITY OF SFM

We benchmarked MSAFlow against MDLM-based decoders under two architectural choices (DiT vs. pretrained ESM) and two conditioning strategies (AdaLN vs. cross-attention). MDLM is a discrete-state mask diffusion model that achieves state-of-the-art performance in natural language generation and provides a strong non–flow-based baseline. Even when warm-started with pretrained ESM weights, MDLM variants remain substantially weaker than our SFM formulation.

Table 13: Ablation on the choice of discrete FM models for decoder

| Model | pLDDT | TM-score |
|---|---|---|
| MSAFlow–MDLM (ESM + cross-attn) | 82.5 | 0.83 |
| MSAFlow–MDLM (DiT + AdaLN) | 79.5 | 0.74 |
| **MSAFlow (SFM, DiT, AdaLN)** | **89.0** | **0.86** |

These results highlight two conclusions: (1) our generative framework is compatible with multiple discrete modeling paradigms, including MDLM; (2) SFM remains decisively superior, validating continuous-state discrete flow-matching with proper manifold geometry as effective mechanism for modeling evolutionary sequence distributions.

### 6.9.2 POSITION-WISE VS. GLOBAL ADALN CONDITIONING

To assess the importance of spatially resolved conditioning, we compared global AdaLN against our proposed position-wise AdaLN. Global conditioning applies a single scale–shift pair across all positions, whereas position-wise AdaLN modulates each residue independently. Our ablations show that global conditioning is far too coarse to capture residue-level evolutionary constraints.

Table 14: Ablation on position-wise vs. global AdaLN conditioning

| Model | pLDDT | TM-score |
|---|---|---|
| MSAFlow (DiT + global AdaLN) | 42.9 | 0.32 |
| **MSAFlow (DiT + position-wise AdaLN)** | **89.0** | **0.86** |

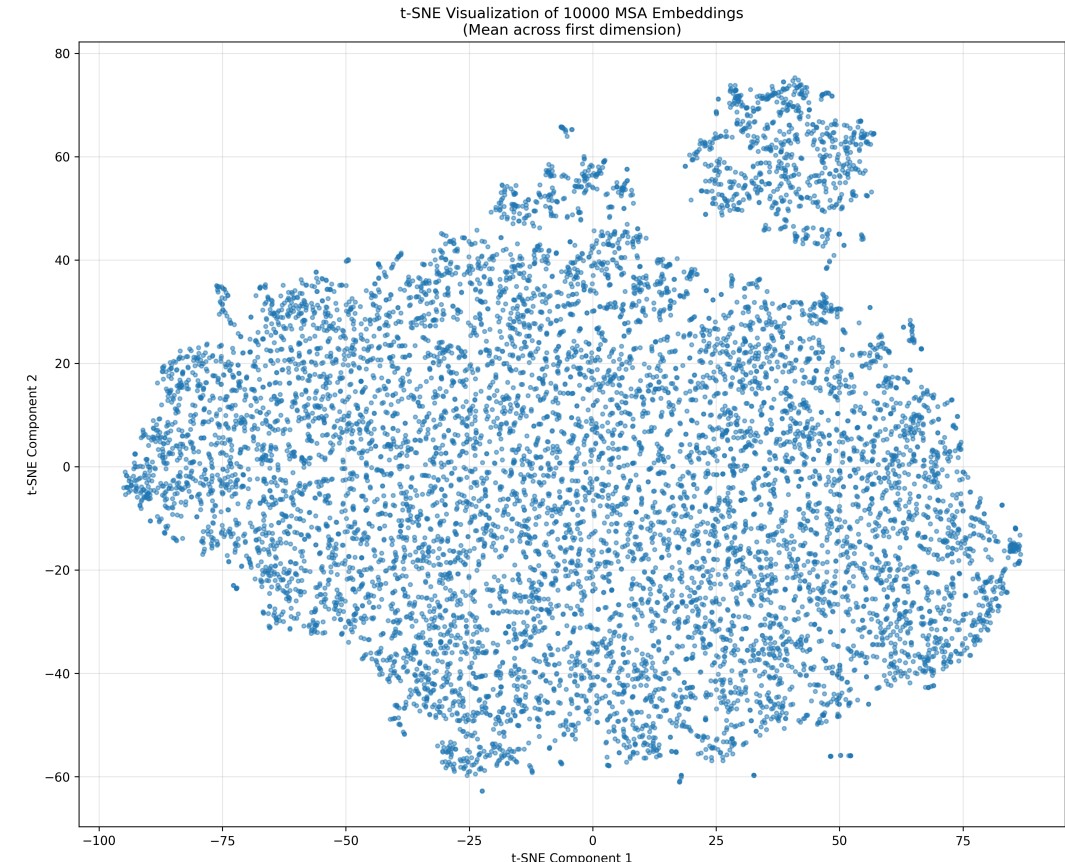

Figure 6: t-SNE projection of 10,000 latent MSA embeddings (mean across the first dimension).

The dramatic degradation under global AdaLN confirms that fine-grained, residue-level conditioning is essential for representing evolutionary constraints, and it empirically validates position-wise AdaLN as a crucial architectural component of MSAFlow.

### 6.9.3 DEPENDENCE ON PRETRAINED ENCODERS.

To quantify how much performance stems from the pretrained encoder versus the flow-matching framework, we conducted an ablation study replacing the AF3 encoder with a smaller, pretrained MSAPairformer encoder (111M params) on a smaller set of training data (100k MSAs).

| Model | pLDDT | TM score |
|---|---|---|
| No MSA | 47 | 0.33 |
| MSAPairformer Embeddings (5 epochs) | 70 | 0.53 |
| AlphaFold3 Embeddings (5 epochs) | 85 | 0.79 |

Table 15: Ablation comparing different pretrained encoders used within MSAFlow.

### 6.10 T-SNE VISUALIZATION OF LATENT MSA EMBEDDINGS

To further support the interpretability of our latent space, we include a t-SNE projection of 10,000 latent MSA embeddings (Fig. 6). The visualization reveals diffuse global structure with numerous small, locally coherent clusters, consistent with the hypothesis that the model organizes sequences according to shared evolutionary or functional patterns. While t-SNE is inherently qualitative, this emergent clustering aligns with prior findings that deep protein embeddings naturally reflect structural

and phylogenetic constraints Alley et al. (2019); Marquet et al. (2022), even in the absence of explicit MSA conditioning. Combined with our strong performance across reconstruction, augmentation, and enzyme design tasks, these patterns suggest that MSAFlow's latent representations meaningfully compress evolutionary variability into a compact and biologically informative manifold.

## 7  USAGE OF LANGUAGE MODELS

We use large language model (LLM) to aid in the preparation of this manuscript. Its use was limited to editorial tasks, including proofreading for typographical errors, correcting grammar, and improving the clarity and readability of the text.

