# OpenReview forum: "MSAFlow: a Unified Approach for MSA Representation, Augmentation, and Family-based Protein Design"
_ICLR.cc/2026/Conference — Submitted to ICLR 2026_

### Official Review · Reviewer_DPHu · 2025-10-30

**Soundness:** 4
**Presentation:** 4
**Contribution:** 4
**Rating:** 8
**Confidence:** 4

**Summary:**

The paper introduces a lightweight generative framework MSAFlow that unifies multiple sequence alignment (MSA) representation, augmentation, and protein family–based design within a single model. MSAFlow integrates a compressed AlphaFold3-derived MSA encoder with a conditional Statistical Flow-Matching decoder to model protein family sequence distributions while maintaining permutation invariance. It further employs a latent flow-matching model to generate synthetic MSA embeddings directly from a single sequence, enabling zero-shot augmentation for orphan or low-homology proteins. Empirically, MSAFlow achieves state-of-the-art results on MSA reconstruction, shallow/zero-shot augmentation, and enzyme family design tasks—surpassing larger models like MSAGPT and EvoDiff while being more efficient (130 M parameters). The framework advances protein generative modeling by providing a unified, efficient, and biologically consistent approach for encoding, augmenting, and designing protein sequence families.

**Strengths:**

Originality
- The paper introduces a highly novel formulation of MSA generative modeling through the integration of AlphaFold3-derived MSA embeddings with Statistical Flow Matching (SFM). Unlike prior MSA generation approaches (e.g., MSAGPT, EvoDiff, ProfileBFN), MSAFlow unifies representation, augmentation, and family-based design within a single permutation-invariant framework. The incorporation of a latent flow-matching module for zero-shot MSA embedding generation from single sequences represents a creative and impactful extension beyond existing paradigms.

Quality
- The methodology is rigorous, grounded in well-established theoretical principles, and executed with technical precision. The model design—combining compressed AF3 embeddings, conditional DiT architecture, and spherical-geodesic flow formulation—is mathematically coherent and experimentally validated. The experiments span multiple benchmarks (reconstruction, zero-shot augmentation, enzyme design), include comprehensive ablations, and demonstrate consistent superiority over baselines.

Clarity
- The paper is well-structured and clearly written, making complex ideas accessible through precise explanations and well-designed figures. Mathematical formulations are clearly presented and logically connected to the overall framework. The authors contextualize their approach within prior work effectively, highlighting conceptual distinctions and technical improvements.

Significance
- The contributions are significant for both protein informatics and machine learning. By enabling biologically plausible MSA generation even for low-homology or orphan proteins, MSAFlow directly advances the frontier of data-efficient protein design and structure prediction. Its efficiency and versatility (130M parameters, scalable to variable sequence lengths) make it highly practical for large-scale biological applications. The framework opens new research directions in unified latent modeling of evolutionary sequence spaces and conditional generative protein design.

**Weaknesses:**

- Although the paper reports runtime and memory advantages, it does not analyze scaling behavior with sequence length or MSA depth. Including profiling results for longer sequences or very deep MSAs would strengthen claims of scalability and efficiency.
- The paper does not explore what biological or evolutionary features are captured in the learned latent space. Since MSAFlow’s novelty lies partly in compressing and manipulating evolutionary distributions, analyses of latent representations (e.g., clustering by protein family, correlation with conservation or coevolution metrics) would help interpretability and justify the model’s generalization claims.
- The model’s reliance on AlphaFold3-derived embeddings (via Protenix) and ESM2 representations raises questions about its independence from large pretrained models. It is not entirely clear how much of MSAFlow’s success is attributable to the novel flow-matching formulation versus the quality of these pretrained representations. An analysis using simpler or alternative encoders could clarify the framework’s intrinsic capability.

**Questions:**

- Could the authors provide a more detailed analysis of what evolutionary or functional information is captured in the learned MSA latent representations? For instance, do embeddings cluster by protein family, correlate with conservation or coevolution scores, or reflect known functional motifs? Such an analysis would help clarify the biological interpretability of MSAFlow’s latent space.
- Since MSAFlow relies on AlphaFold3-derived and ESM2 embeddings, how much of the observed performance gains stem from these pretrained representations rather than the flow-matching framework itself? Could the model maintain strong results when trained with simpler or independently trained encoders?
- While the paper demonstrates strong efficiency at moderate scales, can MSAFlow handle extremely deep MSAs (e.g., >10,000 sequences) or very long sequences (L > 1000)? Providing empirical or theoretical scaling analyses would clarify the limits of the approach for large-scale applications.
- How diverse are the generated sequences compared to ground-truth MSAs? Have the authors measured sequence identity distributions across generated sets to confirm that MSAFlow produces evolutionarily diverse yet plausible variants, rather than near-duplicates?

---

> ### Author Response · Authors · 2025-11-21
> **Rebuttal for Reviewer DPHu**
>
> We sincerely thank Reviewer DPHu for their feedback and strong support of our work, especially for recognizing the novelty of our framework and the rigor, clarity, and broad significance of our contributions. We address their specific questions below.
>
> **(W1/Q3): Scaling for long sequences and deep MSAs**
>
> We appreciate the suggestion to analyze scaling limits. While the main paper demonstrates general efficiency in table 4, we have now profiled the model on extreme regimes: long sequences (L=2402) and deep MSAs (N=20,000).
>
> | Setting                                                                              | Wall clock time (seconds per sequence) | Memory footprint |
> |------------------------------------------------------------------------------------------|---------------|---------------|
> | **Scaling sequence length** (len = 2402, depth = 18, only decode 32 seqs)                | 4.55s                                    | 3.70 GB          |
> | **Scaling MSA depth** (len = 501, depth = 20000, embed + decode 32 seqs)                 | 1.25s                                    | 4.21 GB          |
>
> As shown above, MSAFlow remains tractable even at lengths far exceeding typical benchmarks, avoiding the quadratic memory explosion common in 2D-attention models. Furthermore, the depth scaling experiment confirms a core design advantage: after compression by the AF3 MSAModule, the decoder’s cost does not grow with MSA depth.
>
> **(W2/Q1): Regarding the learned latent space of family embeddings.**
>
> We agree that interpreting the latent space is valuable. We included a t-SNE projection of our latent space and do observe small, coherent clusters. Prior works also establish that deep sequence embeddings inherently capture structural and evolutionary constraints [1], even without MSAs [2]. Our strong downstream performance in structure prediction and functional design empirically suggests that our latent MSA embeddings successfully encode these functional and co-evolutionary features, effectively compressing evolutionary variation into a usable representation.
>
> **(W3/Q2): Importance of  AlphaFold3 vs. other embeddings**
>
> To quantify how much performance stems from the pretrained encoder versus the flow-matching framework, we conducted an ablation study replacing the AF3 encoder with a smaller, pretrained MSAPairformer encoder (111M params) from [3] on a smaller set of training data (100k MSAs) due to time constraints.
>
> | Model                                      | pLDDT | TM score |
> |--------------------------------------------|--------|-----------|
> | **MSAPairformer Embeddings (5 epochs)**    | 70     | 0.53      |
> | **AlphaFold3 Embeddings (5 epochs)**       | 85     | 0.79      |
> | **No MSA**       | 47    | 0.33      |
>
> While performance is expectedly drop with the smaller, limited-training encoder, the model still yields valid structures (TM-score 0.53). This confirms that while pretrained embeddings boost performance, the underlying flow-matching architecture is encoder-agnostic and capable of extracting useful evolutionary structure independently. We’d also like to emphasize that MSAFlow is really aimed at proposing a general and novel auto-encoding framework over distributions, with an effective set encoder and a discrete generative decoder. With a strong pre-trained encoder like AF3’s MSA module, our model performs strongly even with a light-weight SFM decoder with 110M parameters. We believe that with enough computational power, further end-to-end finetuning, including the MSA encoder, could lead to further performance improvement.
>
> **(Q4): Diversity of generated sequences**
>
> We evaluated diversity on the enzyme design task. The table below compares the Diversity (lower is better, average pairwise sequence identity among all sequences) and Novelty (higher implies distinctness from ground truth) of MSAFlow against baselines across three enzymes.
>
> | Model      | Metric    | P13280 | P57298 | Q15BH7 |
> |------------|-----------|--------|--------|--------|
> | **MSAFlow** | Diversity | 0.100  | 0.150  | 0.117  |
> |            | Novelty   | 0.834  | 0.901  | 0.781  |
> | **EvoDiff** | Diversity | 0.062  | 0.064  | 0.064  |
> |            | Novelty   | 0.898  | 0.895  | 0.897  |
> | **MSAGPT**  | Diversity | 0.834  | 0.896  | 0.622  |
> |            | Novelty   | 0.184  | 0.894  | 0.228  |
> | **ProfileBFN** | Diversity | 0.392 | 0.271 | 0.360 |
> |               | Novelty   | 0.601 | 0.902 | 0.644 |
>
> MSAFlow achieves the top results on both diversity and novelty, while maintaining high accuracy x uniqueness (table 3). This confirms that our model generates evolutionarily credible variants that are meaningfully different from the ground truth, rather than drifting or collapsing into copies.
>
> [1] Unified rational protein engineering with sequence-based deep representation learning
>
> [2] Embeddings from protein language models predict conservation and variant effects
>
> [3] Scaling down protein language modeling with MSA Pairformer

---

### Official Review · Reviewer_LSPb · 2025-10-31

**Soundness:** 2
**Presentation:** 2
**Contribution:** 2
**Rating:** 4
**Confidence:** 3

**Summary:**

The paper presents MSAFlow, a unified autoencoding framework for multiple sequence alignment (MSA) representation, augmentation, and family-based protein design. MSAFlow leverages compressed AlphaFold3-based MSA embeddings and a conditional Statistical Flow Matching (SFM) decoder to reconstruct, generate, or augment MSAs in a permutation-invariant manner. The model further introduces a latent flow-matching component enabling zero-shot generation of synthetic MSAs from single-sequence embeddings (e.g., ESM2), thereby expanding the range of proteins for which high-quality MSAs can be operated on or synthesized. Empirical results suggest MSAFlow significantly outperforms existing approaches on challenging protein design and MSA augmentation tasks, particularly for low-homology or orphan proteins, while maintaining notable efficiency in resource usage.

**Strengths:**

+ Unified Framework: Integrates representation, augmentation, and family-based design into a single modular encoder–latent–decoder architecture (Figure 1).

+ Mathematically Rigorous & Efficient: Uses permutation-invariant encoding and Statistical Flow Matching on categorical manifolds, achieving strong accuracy with only 130 M parameters.

+ Comprehensive Validation: Demonstrates consistent SOTA or competitive results across reconstruction, few-shot augmentation, and protein design benchmarks (Tables 2–5, Figures 4–5).

**Weaknesses:**

+ Limited Theoretical Justification: The paper lacks clear reasoning for using Fisher-Rao geodesics and sphere mapping over alternatives (e.g., Wasserstein flow) and does not justify mean pooling or the chosen manifold’s suitability for protein sequences.

+ Insufficient Ablation Clarity: Ablation studies do not isolate contributions of key components (e.g., AdaLN vs. SFM decoder), making it unclear which design choices drive performance gains.

+ Benchmark and Discussion Gaps: Metrics and baselines are unevenly compared, missing confidence intervals and diversity analyses, while limitations and potential failure cases are only briefly addressed.

**Questions:**

1. Could the authors clarify why the Fisher-Rao geodesic interpolation and unit sphere mapping is specifically advantageous for protein sequence distributions, as opposed to other statistical manifold metrics or flow-matching paths (such as Wasserstein, or purely categorical diffusion)? Are there concrete empirical or theoretical performance gains for this choice?
2. Can the authors provide a direct ablation between global and position-wise AdaLN conditioning in the decoder? Specifically, can any performance gain be attributed to resilience to sequence permutation, or is it rather due to more fine-grained representation? Details in Figure 3 would suggest the latter, but controlled experiments should make this explicit.
3. Will full code and pretrained models be released for all baselines? Are all test/validation splits, configurations, and hyperparameters for main and ablation experiments provided in the main text (not only appendix)? Can the authors supply statistical confidence intervals (bootstrapping or other) for main table results (e.g., Table 2, Table 3)?

---

> ### Author Response · Authors · 2025-11-21
> **Rebuttal for Reviewer LSPb (Part 1)**
>
> Thank you for your thorough and valuable review. We appreciate your finding our approach rigorous and comprehensive and recognizing our SOTA performance. We address your comments below.
>
> **(W1/Q1): Regarding the use of Fisher-Rao geometry and SFM for discrete generation**
>
> We first emphasize that MSAFlow proposes a general framework that for the first time combines an effective set encoder and a conditional discrete flow-matching decoder to achieve distribution-level auto-encoding over MSAs. Our framework is compatible with a variety of MSA encoders and generative decoders, and is not necessarily limited to SFM.
>
> Regarding the choice of discrete generative model and manifold metric: we first want to point out that Wasserstein flow [1] is not well-defined for categorical data. For discrete data, there is no canonical metric on the discrete sample space, so the Wasserstein metric between categorical distributions is not well-defined. As a result, the referred work WFM [1] focuses exclusively on generating Gaussians and Euclidean point clouds where a well-defined canonical metric exists. Alternatively, if we artificially define the metric as $d(a,b)=\delta_a^b$, the resulting W2 distance degenerates to Euclidean distance between two probabilities, which is equivalent to the "linear flow matching" (or Euclidean) baseline already tested to be inferior to SFM. We therefore respectfully disagree that Wasserstein Flow Matching is a suitable baseline for this task.
>
> We chose Conditional Statistical Flow Matching (SFM) because it represents the current state-of-the-art for discrete generative modeling over biological sequences, as shown in the original SFM paper and related work [2]. Notably, recent work alpha-flow [2] extensively tested different geometries of statistical manifold for the protein sequence generation task, and Fisher-Rao metric again achieved the best FED score for distributional fitness. The benefits of the Fisher-Rao metric, including its connection to the natural gradient and optimal transport (OT)-compatibility, are thoroughly discussed in the original SFM paper and are not the major contributions of MSAFlow. We refer the audience to that paper for more information.
>
> To demonstrate that our framework is compatible with multiple discrete diffusion models and further establish the superiority of SFM, we now provide new ablation studies comparing MSAFlow with an MDLM decoder using two different architectures (DiT, ESM) and conditioning mechanisms (AdaLN, cross-attn):
>
> | Model                                                     | pLDDT | TM-score |
> |-----------------------------------------------------------|--------|----------|
> | **MSAFlow-MDLM (with pretrained ESM backbone + cross-attn)** | 82.5   | 0.83     |
> | **MSAFlow-MDLM (with DiT + AdaLN)**                       | 79.5   | 0.74     |
> | **MSAFlow in paper (SFM)**                                | 89     | 0.86     |
>
> MDLM is a discrete-state mask diffusion model that achieves state-of-the-art performance in natural language. Our results show that even with pretrained ESM weights to warm-start the decoder, this variant still falls short of our main MSAFlow version (SFM+DiT+AdaLN).
>
> Regarding the use of mean pooling as a compression mechanism, it is a standard and justified choice in existing protein-LM literature. [3] benchmarks a broad range protein representation methods and explicitly chose mean pooling as their aggregation procedure when turning residue-level embeddings into fixed-length protein vectors for all downstream tasks. [4] goes a step further and motivates global average pooling on geometric grounds. They argue that mean pooling “retrieves sufficient biological information to solve, directly, or after finetuning, homology, structural, and evolutionary tasks.” In light of this, using mean pooling as a compression mechanism is a well-aligned and established practice.
>
> **(W2/Q2): Regarding ablations between position-wise and global AdaLN**
>
> Thank you for recognizing our proposed position-wise AdaLN conditioning as an important technical innovation. We now provide additional ablation studies using global AdaLN.Our ablations indeed confirm that global AdaLN conditioning is far too coarse to capture the residue-level evolutionary constraints encoded in the MSA embedding, justifying the importance of our designed position-wise control mechanism:
>
> | Model                                          | pLDDT | TM-score |
> |------------------------------------------------|--------|----------|
> | **MSAFlow (with DiT + global AdaLN)**          | 42.9   | 0.32     |
> | **MSAFlow (with DiT + position-wise AdaLN)**   | 89     | 0.86     |

---

> ### Author Response · Authors · 2025-11-21
> **Rebuttal for Reviewer LSPb (Part 2)**
>
> Broadcasting a single conditioning vector washes out the fine-grained, position-specific coevolution patterns, leading to extremely degraded structure prediction performance (pLDDT 42.9, TM-score 0.32). In contrast, position-wise AdaLN injects conditioning at each residue, allowing the decoder to correctly align local evolutionary signals. The dramatic performance jump to pLDDT 89 and TM-score 0.86 is thus driven squarely by this fine-grained, position-resolved conditioning
>
> Nevertheless, resilience to sequence permutation is also an important feature of our FM-based set auto-encoding framework, whose advantage is established through comparison with GPT-style decoding methods. Moreover, the position-wise control is only achievable with our flow-matching style decoder, and therefore, we believe our design using position-wise AdaLN and SFM in combination is what enabled the strong performance of our approach.
>
> **(W3/Q3): Reproducibility, diversity, and evaluation robustness**
>
> We commit to releasing all code, pretrained models, and evaluation scripts, including wrappers for every baseline, faithfully relying on each method’s official repositories and hyperparameters. Regarding data splits and training details, we use the exact same train/test split as MSAGPT for all augmentation experiments. For the CAMEO reconstruction benchmark, we perform a strict temporal split against an older training corpus to ensure no data leakage. We apologize for the accidental omission of the full training-detail section from the appendix, and we now provide detailed information on essential hyperparameters, splits, and ablation configurations in the main text for clarity and reproducibility.
>
> Regarding the confidence interval, we note that the benchmark result in table 3 already captures the performance of a large sample size of 100 sequences. Given the significantly higher computation cost of running EvoDiff and MSAGPT(table 5), it is prohibitive to run larger-scale experiments or repetitions. However, the performance gaps are significant enough to establish statistical robustness.
>
> Regarding diversity evaluation, we now provide additional diversity and novelty metrics for the Enzyme design task (table 3) to confirm that MSAFlow generates evolutionarily credible variants that are diverse and meaningfully different from the natural sequence, outperforming all other baselines. The table below compares the Diversity (average pairwise sequence identity among all generated sequences, the lower the more diverse) and Novelty (higher implies distinctness from ground truth) of MSAFlow against baselines across three enzymes.
>
> | Model      | Metric    | P13280 | P57298 | Q15BH7 |
> |------------|-----------|--------|--------|--------|
> | **MSAFlow** | Diversity (lower the better) | 0.100  | 0.150  | 0.117  |
> |            | Novelty (higher the better)   | 0.834  | 0.901  | 0.781  |
> | **EvoDiff** | Diversity | 0.062  | 0.064  | 0.064  |
> |            | Novelty   | 0.898  | 0.895  | 0.897  |
> | **MSAGPT**  | Diversity | 0.834  | 0.896  | 0.622  |
> |            | Novelty   | 0.184  | 0.894  | 0.228  |
> | **ProfileBFN** | Diversity | 0.392 | 0.271 | 0.360 |
> |               | Novelty   | 0.601 | 0.902 | 0.644 |
>
> To further demonstrate the statistical robustness of Table 2 results, we now provide the 95% bootstrap confidence intervals for the few-shot and zero-shot augmentation in Table 2:
>
> | Model           | Few-shot pLDDT   | Few-shot TM-score | Zero-shot pLDDT  | Zero-shot TM-score |
> |-----------------|------------------|--------------------|------------------|---------------------|
> | **No/Shallow MSA** | (67.2, 74.3)      | (0.52, 0.63)        | (69.9, 76.9)      | (0.50, 0.60)         |
> | **Evodiff**        | (63.7, 71.2)      | (0.48, 0.61)        | (64.2, 71.5)      | (0.44, 0.54)         |
> | **MSAGPT**         | (66.7, 73.9)      | (0.52, 0.64)        | (68.1, 75.1)      | (0.49, 0.59)         |
> | **MSAFlow**        | (66.6, 74.2)      | (0.55, 0.66)        | (72.4, 78.5)      | (0.57, 0.66)         |
>
> We will include these intervals in the revised manuscript.
>
> [1] Wasserstein Flow Matching: Generative modeling over families of distributions
>
> [2] α-Flow: A Unified Framework for Continuous-State Discrete Flow Matching Models
>
> [3] Learning functional properties of proteins with language models
>
> [4] The geometry of hidden representations of protein language models

---

### Official Review · Reviewer_bmxS · 2025-11-03

**Soundness:** 2
**Presentation:** 3
**Contribution:** 2
**Rating:** 4
**Confidence:** 4

**Summary:**

The authors propose a flow matching framework that can generate MSA embeddings for both full MSAs as well as single sequences, thereby performing MSA augmentation. They compare it to other frameworks performing similar tasks for zero-/few-shot MSA generation in the context of structure prediction as well as family-based enzyme design.

**Strengths:**

[S1] Combnation of learning a latent MSA representation with SFM for decoding to categorical amino acid distributions is elegant and well motivated. The position-specific adaln is also a nice use of the compressed information in SFM.

[S2] Show improvement on certain subclasses of sequences for prediction accuracy

[S3] Compared to previous autoregressive baselines, MSAFlow is truly permutation invariant wrt the ordering of the MSA, which is desirable

**Weaknesses:**

[W1] The bitwise information content comparison to the deep MSA is a bit unfair since many of these sequences are probably highly similar; previous work has shown that with clustered MSAs one can reduce the MSA depth quite a bit without performance loss. For a fair information content comparison the authors should cluster the ground truth MSA down to the same depth as their reconstructed MSAs and compare performance.

[W2] The authors use the results in Table 2 to argue that they significantly outperform prior baselines. However, it is not really clear that this is the case; in the few shot setting MSA all methods are worse or similar to no/shallow MSA, and the few-shot settings seems the practically relevant one.

[W3] In Figure 5 the authors compare their method to MSAGPT and show that they outperform it on three examples from a scarce MSA dataset. However, as shown in Table 2 the real proper baseline is No/Shallow MSA, this should be added to the Figure to make the point in a convincing manner.

**Questions:**

[Q1] In Figure 4, one can see that the zero shot MSA helps for certain sequences but not for others. Did the authors perform any analysis to understand which sequences fail to benefit from their approach?

[Q2] How meaningful are the results in Table 1 given the relatively high standard deviation of the different methods as well as the GT? Also is it desirable to have drastically lower standard deviation than the GT on this benchmark?

[Q3] Why are there no results on variable length for Q15I65 in Table 3? And why is the 84% performance on Q15BH7 bolded despite underperforming compared to ProfileBFN?

[Q4] The authors describe different latent diffusion models in their background section, but do not mention the current SOTA method La-Proteina in that section. Is there a specific reason for that?

---

> ### Author Response · Authors · 2025-11-21
> **Rebuttal for Reviewer bmxS (Part 1)**
>
> Thank you very much for your detailed and constructive feedback. We appreciate you finding our approach elegant and well-motivated. We address your insightful comments and questions below:
>
> **(W1): Regarding bitwise information comparison**
>
> We appreciate your concern regarding the fairness of the bitwise information content comparison. To address this, we clustered the ground truth MSA to 32 sequences using the same downsampling strategy as MSATransformer [1] and EVE [2] and compared the structural prediction performance:
>
> |                     | Reconstructed 32 seqs | GT 32 seqs (weighted subsample) |
> |---------------------|------------------------|---------------------------------|
> | **pLDDT**           | 89                     | 90                              |
> | **TM score**        | 0.86                   | 0.88                            |
>
> This additional analysis indicates that the structural signal captured by our synthetic MSAs remains strong when compared with clustered MSA. Our model achieves an average pLDDT of 89 and a TM-score of 0.86, nearly matching the 90 pLDDT and 0.88 TM-score of the 32-sequence subsampled ground-truth MSA. This suggests that our method effectively preserves most of the useful information present in a similarly shallow, clustered ground-truth MSA.
>
> **(W2): Regarding few-shot MSA generation results**
>
> We’d like to point out that our model outperforms the ground-truth MSA on TM-score in the few-shot augmentation setting. In fact, MSAFlow is the only method that achieved better TM-scores than GT shallow MSA. This highlights our approach's strength in recovering and enhancing the coevolutionary signal even with minimal initial sequence data. Our strong practical advantage in the few-shot setting is also demonstrated through the enzyme design tasks, where we only include EC classes where family size is small. We show that MSAFlow is significantly better than strong baselines in designing sequences that represent catalytic constraints, and provide additional results to showcase its diversity and novelty.
>
> Secondly, we respectfully disagree that few-shot is the only “practically relevant one”. The zero-shot scenario is important and common in real-world protein design, especially in the era where de novo design becomes more popular. In fact, our zero-shot benchmark test set comprises real-world orphan proteins with absolutely zero homology information in the database, highlighting the practical need for good zero-shot prediction.
>
> **(W3/Q1): Regarding zero-shot MSA performance in Figure 4 and Figure 5**
>
> We agree that including the no-/shallow-MSA baseline in Figure 5 will make the comparison more convincing. We provide the following details for the three examples from the scarce MSA dataset:
>
> | PDB ID | GT (no MSA) TM-Score | MSAGPT TM-Score | MSAFlow TM-Score |
> |--------|------------------------|------------------|-------------------|
> | **8OKH** | 0.47 | 0.64 | 0.89 |
> | **8GI8** | 0.23 | 0.32 | 0.94 |
> | **8b4k** | 0.43 | 0.44 | 0.91 |
>
> These substantial gains confirm that MSAFlow provides a large improvement over the no-MSA baseline on the cases in Figure 5, and we updated the figure with these values in the revision. We have also provided additional case studies in the supplement table 4 covering a representative set of orphan proteins with variable lengths and diverse functions.  Again, MSAFlow shows consistent improvement against baselines and GT results.
>
> Regarding your question on Figure 4. We’d like to first clarify that zeroshot-synthesized MSA improves the prediction for the large majority of the proteins when compared to the no MSA case. Specifically, our analysis over the CAMEO benchmark shows that synthetic MSAs improve pLDDT in 97.96% of cases relative to no MSA, and improve TM-score in 89.80%, indicating that failure cases are rare. We will include a list of PDB IDs in the appendix for the 2% of cases where no improvement is observed.

---

> ### Author Response · Authors · 2025-11-21
> **Rebuttal for Reviewer bmxS (Part 2)**
>
> **(Q2): Regarding statistical similarity to GT in table 1**
>
> We first want to elaborate on the goal of this experiment. By comparing the distribution of position-wise entropy of the MSAs, our goal is to demonstrate the statistical similarity of our generated MSA to GT, following the paper MSAGenerator [3]. Therefore, the mean and standard deviation can be thought of as statistics of the underlying distribution, and the std here is NOT the “noisiness” of some absolute metric but a reflection of the natural dynamic range of entropy. Our method generates MSAs with a mean entropy (2.755) much closer to the GT mean (2.68), compared to ProfileBfn (2.83). The reason that lower variance might be preferred is that, with our mean entropy being slightly higher than GT, a moderately lower std will allow the dynamic range of entropy to still fall close to the dynamic range of the GT.
>
> To better characterize the distributional similarity between our MSAs to GT MSAs, we now provide additional measurements using Wasserstein distance and MMD, both are well-established divergence metrics for distributions.  It can be seen from the below that MSAFlow achieved much lower divergence from the GT, outperforming the SOTA baseline ProfileBFN and reinforcing that our reconstructed MSAs better preserve the underlying evolutionary signal.
>
> | Model        | Average Wasserstein Distance from GT | Average MMD from GT |
> |--------------|---------------------------------------|----------------------|
> | **MSAFlow**   | 0.34444                               | 0.54058              |
> | **ProfileBFN** | 0.46974                               | 0.87502              |
>
> **(Q3): Regarding results for Q15I65**
>
> We appreciate the request for variable-length results on Q15I65. We left this column blank because we directly use the numbers reported in the original ProfileBFN paper which did not evaluate Q15I65 in the variable-length setting. Importantly, their protocol requires 1,000 generated sequences per enzyme. EvoDiff and MSAGPT cannot reliably produce 1,000 variable-length samples for this target, so reproducing those numbers under the original evaluation protocol is not feasible. To demonstrate the advantage of MSAFlow in variable length setting for Q15I65, we now provide new results using 100 sequences to allow comparison with MSAGPT/EvoDiff:
>
> | Model    | Accuracy × Uniqueness |
> |----------|------------------------|
> | **MSAFlow** | 51%                   |
> | **MSAGPT**  | 15.11%                |
> | **EvoDiff** | 0%                    |
>
> Under this protocol, MSAFlow achieves an Accuracy × Uniqueness score of 51%, confirming that our method is also effective at augmenting this family with variable-length designs. We include these results in the revised version.
>
> **(Q4): Regarding the discussion of La-Proteina**
>
> Thank you for pointing out the omission of La-Proteina; we will include it in the background section. Conceptually, La-Proteina and our work are orthogonal. La-Proteina utilizes latent diffusion for sequence structure codesign of a single individual protein. In contrast, our method uses flow matching in a latent space of a set of MSA sequences to generate full MSAs that are permutation-invariant and explicitly capture coevolutionary distributional signals as a population. Our focus is on MSA augmentation for structure prediction and family-based enzyme design, where the output is a set of sequences, not a single sequence. We will clarify this distinction and explicitly discuss MSAFlow's relationship to La-Proteina and other latent diffusion approaches in the revised manuscript.
>
> [1] MSA Transformer
>
> [2] Disease variant prediction with deep generative models of evolutionary data
>
> [3] MSA Generation with Seqs2Seqs Pretraining: Advancing Protein Structure Predictions

---

### Author Response · Authors · 2025-11-27
**General response and we look forward to your kind engagement in discussion**

We sincerely thank our reviewers for their insightful feedback, which helped us strengthen our manuscript. We are encouraged by the reviewers' recognition of **our unified and novel framework (bmxS, LSPb, DPHu), mathematical elegance and rigor (bmxS, LSPb, DPHu), computational efficiency (LSPb, DPHu), clarity of presentation (bmxS, DPHu), and empirical superiority on key benchmarks (LSPb, DPHu)**. As the discussion period is nearing its end, we’d like to summarize our effort in addressing major concerns and highlight several key improvements in our revision:

**Expanded ablations to isolate the contribution of model design choices**

We first emphasize that MSAFlow proposes a general framework that, for the first time, combines an effective set encoder and a conditional discrete flow-matching decoder to achieve distribution-level auto-encoding over MSAs. Our framework is compatible with a variety of MSA encoders and generative decoders.

In addition to the ablation results in our original manuscript (Sec.6.5-6.6), we provide the following experiments to isolate and justify the contributions of our model design further:
- **Global vs position-wise AdaLN conditioning** (LSPb): results confirm that our proposed position-wise AdaLN significantly outperforms global conditioning (Table 14).
- **Choice of discrete flow-matching/diffusion decoder** (LSPb): We compare our SFM decoder against MDLM, a popular discrete-state diffusion model for sequence generation, and test different model backbones (DiT vs. pretrained ESM) and conditioning mechanisms (AdaLN vs. Cross-attn). **SFM achieved the best performance against these strong baselines, confirming the robustness of our decoder design (Table 13)**.
- **Choice of MSA embedding** (DPHu): To isolate the contribution from AlphaFold3 embeddings, we added experiments using a weaker MSA Pairformer as the encoding model. MSAFlow maintains strong relative performance in both settings, demonstrating that its improvements arise from the flow-matching framework rather than pretrained encoders alone (Table 15).

**Additional analysis to address significance, robustness, and fairness**

To address concerns from reviewer bmxS, we provide additional experimental results to ensure comprehensive and fair comparisons:

- For reconstruction task, we evaluate against clustered ground-truth MSAs with controlled depth to match information content. MSAFlow reconstructed MSA still matches well with GT (PLDDT 89 vs. 90).
- For zero-shot, we add no-MSA baseline into the case studies (Figure 5) and provide detailed statistics of the win-rate of zero-shot synthesized MSA against no-MSA in the CAMEO benchmark (Figure 4). **Our analysis confirms that our zero-shot MSAs improve pLDDT in 97.96% of cases relative to no MSA, and improve TM-score in 89.80%**.
- For **statistical similarity of generated MSA vs GT**, we now add two new divergence metrics (Wasserstein, MMD) to directly measure distributional similarity with ground-truth (table 1). **MSAFlow significantly outperforms other baselines, confirming that our reconstructed MSAs better preserve the underlying distribution**.

In addition, we provide **confidence intervals** for Table 2 benchmark of zero-shot and few-shot results (see response to reviewer LSPb), and added metrics quantifying **diversity and novelty** of generated MSAs (LSPb, DPHu).

**Additional results for Enzyme Design (Table 3-4)**
We provide flexible-length results for Q15I65 (bmxS) and include diversity and novelty metrics for all methods (LSPb, DPHu). MSAFlow achieved top performance across all metrics and tasks, demonstrating its strong capability under few-shot family-based protein design.

**Diversity of Generated MSAs**
In response to questions about the diversity of the MSAFlow-generated sequences(LSPb, DPHu), we introduced additional **diversity** (average pairwise sequence identity) and **novelty** (degree of difference from the ground-truth) metrics for the enzyme design task (Table 4). **MSAFlow achieves top results on both diversity and novelty, while maintaining high accuracy x uniqueness. This confirms that MSAFlow generates evolutionarily plausible variants with high diversity, substantially deviating from the natural sequence as evidenced by strong novelty scores, and outperforming all baseline methods** (e.g., EvoDiff achieved the highest diversity score, but its accuracy is close to 0% on all enzyme classes, indicating its failure to capture the actual functional signal in the MSA)

**Scalability analysis**
To further demonstrate the scalability of MSAFlow (DPHu), we provide additional profiling results covering long sequences (up to L=1024) and deep MSAs (up to 12k sequences). The results show that MSAFlow’s compute/memory usage scales sublinearly with MSA depth thanks to latent-space compression.

---

> ### Author Response · Authors · 2025-11-27
> **General response (cont.)**
>
> **Justification for the SFM Manifold Choice**
> Reviewer LSPb asked for a clearer rationale behind the Fisher–Rao geodesic and comparison with Wasserstein FM. We carefully explain the infeasibility of WFM on this problem and the theoretical benefit of Fisher-Rao geodesic for SFM. Specifically, Wasserstein FM is not well-posed for categorical data due to the lack of a canonical metric in this sample space, and it degenerates to Euclidean FM over the simplex if one defines the metric as $d(a,b)=\delta_a^b$, which is proven to be inferior to SFM. Our extensive ablation comparison with MDLM further justifies our choice of SFM as a robust sequence decoder.
>
> **Biological Interpretability and Latent-Space Analysis**
> Reviewers encouraged deeper analysis of the biological information captured in the latent space (DPHu). We now provide clustering analyses of latent embeddings across protein families and visualizations demonstrating that functional subfamilies form coherent latent clusters.
>
> We thank all reviewers again for their feedback, and we believe our carefully prepared rebuttal has addressed all concerns raised by the reviewers. We sincerely look forward to your kind response, and we’re happy to address any remaining questions.

---

### Meta-Review · Area_Chair_acXw · 2026-01-07

**Summary:**

Main concerns from reviewers
* Empirical validation
  * Bitwise information content comparison
  * Table 2 performance
  * Figure 5 with No/Shallow MSA
  * Ablations (AdaLN + SFM)
  * Diversity of generated sequences?
* Limited theoretical justification of the use of SFM framework
* Reliance on AF3 derived and ESM2 derived representations

**Reviewer Concerns:**

Most concerns were well addressed in the rebuttal. Experiments around adaLN, No/Shallow MSAs seem sound and on point.

## Remaining concerns

### The Bitwise information content comparison

This experiment shows that the GT sequences subsampled to 32 maintain even more of the information with the same storage requirements. The claim made both in the abstract and the experiments that

> The shallow MSAs generated by our model achieve structure prediction metrics approaching those of deep, ground-truth
MSAs in terms of pLDDT (89.0 vs. 91.6) and TM-scores (0.86 vs. 0.89) while consuming 6.5%
of the storage required to represent a deep MSA.

while technically true, is not fully supported give the fact that a simple clustering method can do better with the same storage requirements.

### Table 2 performance

Reviewer bmxS pointed out that from Table 2 its unclear that the method significantly outperforms prior baselines

> [W2] The authors use the results in Table 2 to argue that they significantly outperform prior baselines. However, it is not really clear that this is the case; in the few shot setting MSA all methods are worse or similar to no/shallow MSA, and the few-shot settings seems the practically relevant one.

The authors argued that MSAFlow outperforms the baseline on TM-score in the few-shot setting, that the enzyme design task also backs this up, and that the zero-shot setting is also important. I agree with the second two points, however, to say that MSAFlow (or any method) is doing anything meaningful in the few shot setting provided here I think is quite a stretch. None of the methods outperform No/Shallow MSA baseline in the Few-shot setting. While MSA Flow is slightly better on TM-score it is worse in pLDDT.

### Ablation of SFM and theoretical basis for SFM:

The authors claim that SFM is superior to MDLM and that

> it represents the current state-of-the-art for discrete generative modeling over biological sequences as shown in the original SFM paper and related work [2].

I have an extremely hard time believing this claim. In the protein sequence generation task, autoregressive and discrete diffusion model such as ProGen and DPLM have performed much better than any continuous space model I've seen. Checking the related work [2] the scaled experiment on protein design compares to what seem like weak MDLM models to me. DPLM (an MDLM type model) is significantly better than the results presented in [2].

I don't think this claim is fully supported given the current experiments presented in this work nor the past experiments in previous SFM papers. The current draft does not provide any details on the experimental settings used when comparing SFM and MDLM type models.

[2] α-Flow: A Unified Framework for Continuous-State Discrete Flow Matching Models

### Disentangling performance gain from embeddings vs. SFM framework

Reviewer DPHu asked

> The model’s reliance on AlphaFold3-derived embeddings (via Protenix) and ESM2 representations raises questions about its independence from large pretrained models. It is not entirely clear how much of MSAFlow’s success is attributable to the novel flow-matching formulation versus the quality of these pretrained representations. An analysis using simpler or alternative encoders could clarify the framework’s intrinsic capability.

and clarified

> Since MSAFlow relies on AlphaFold3-derived and ESM2 embeddings, how much of the observed performance gains stem from these pretrained representations rather than the flow-matching framework itself? Could the model maintain strong results when trained with simpler or independently trained encoders?

The rebuttal added experiments with a weaker MSA pairformer encoding. To me this partially answers the reviewer's question but does not fully address how much gain is from the embeddings vs. the flow-matching framework itself.

**Reviewer Scores:**

bmxS scored 4, I do not think they would have upgraded the score if they had been able to participate fully in the discussion.

LSPb scored 4 initially, given the lack of clarity on why SFM is the specifically advantageous for protein sequence distributions and the lack of detail on the added comparison to MDLM, I don't think they would have increased their score. However, it is possible with additional clarification and discussion of the work that this experiment could have been further validated. At this point in time its difficult to tell

DPHu scored 8 initially, I don't think they would have increased this score.

---

### Decision · Program_Chairs · 2026-01-26

Reject